# Obtaining and Characterization of Highly Crystalline Recycled Graphites from Different Types of Spent Batteries

**DOI:** 10.3390/ma15093246

**Published:** 2022-04-30

**Authors:** Lorena Alcaraz, Carlos Díaz-Guerra, Joaquín Calbet, María Luisa López, Félix A. López

**Affiliations:** 1Centro Nacional de Investigaciones Metalúrgicas (CENIM), Consejo Superior de Investigaciones Científicas (CSIC), Av. Gregorio del Amo 8, 28040 Madrid, Spain; f.lopez@csic.es; 2Departamento de Física de Materiales, Facultad de Ciencias Físicas, Universidad Complutense de Madrid, Plaza de Ciencias 1, 28040 Madrid, Spain; cdiazgue@fis.ucm.es; 3Departamento de Química Inorgánica, Facultad de Ciencias Químicas, Universidad Complutense de Madrid, Ciudad Universitaria s/n, 28040 Madrid, Spain; jocalbet@ucm.es (J.C.); marisal@ucm.es (M.L.L.)

**Keywords:** recycled graphite, high-quality graphite, spent batteries, acidic leaching

## Abstract

Spent batteries recycling is an important way to obtain low-cost graphite. Nevertheless, the obtaining of crystalline graphite with a rather low density of defects is required for many applications. In the present work, high-quality graphites have been obtained from different kinds of spent batteries. Black masses from spent alkaline batteries (batteries black masses, BBM), and lithium-ion batteries from smartphones (smartphone black masses, SBM) and electric and/or hybrid vehicles (lithium-ion black masses, LBM) were used as starting materials. A hydrometallurgical process was then used to obtain recycled graphites by acidic leaching. Different leaching conditions were used depending on the type of the initial black mass. The final solids were characterized by a wide set of complementary techniques. The performance as Li ion batteries anode of the sample with better structural quality was assessed.

## 1. Introduction

Circular economy is a regenerative approach to reduce the waste generated and guarantee the eco-sustainability of post-use products. When a product reaches the end of its life, it is reused to generate high added-value materials, which reduces both the need for primary materials and waste production [1,2]. This explains the growing interest among the scientific community to develop and optimize different methods for the recycling and the reuse of waste, especially hazardous waste which requires an adequate management.

Batteries are currently used in many electronic devices, including electric and hybrid vehicles [3,4]. Due to the exponentially increasing number of spent batteries recently generated, the recycling of batteries components has aroused global attention [5]. Spent batteries contain heavy metals which may seep out, negatively affecting the environment and human health [6,7]. The leaching process has successfully been described as a method to dissolve metals present in the black mass, a common waste generated in the recycling of batteries.

Usually, a preliminary phase is carried out for spent batteries preparation (sorting, discharging, and dismantling). Subsequently, spent batteries are pre-treated to generate the active valuable materials through different processes (thermal, mechanical, physical, chemical, or mechano-chemical). Thus, a metals-enriched fraction (called black mass) is obtained.

Previous investigations showed that it is possible to obtain highly pure materials from black masses. Regarding the recycling of the anode material, some investigations have previously reported the recovery of high-quality graphite using mechanical methods [8,9] and/or hydrometallurgical processes [10,11] from spent lithium-ion batteries (LIBs). In fact, after the leaching process a large amount of insoluble residues remain, mainly composed of carbon. This carbonaceous residue is an excellent precursor which can be turned into a high added-value material such as graphite, which is used in a great variety of electrochemical applications due to its electrical and thermal conductivity, inertness and resistance. Moreover, natural graphite is considered as a critical raw material by the European Union (EU) since 2011 [12]. In fact, the Critical Raw Materials list has been continuously revised for years and natural graphite remains on such list, suggesting that the threat to future supply is severe and long-term [13]. Actually, the EU predicts a four-fold increase of the demand of graphite by 2030 and a 15-fold increase by 2050 [14]. As an interesting alternative, one of the initiatives fostered by the EU is to secure the supply of these critical materials through recycling strategies. In addition, it was reported that the demand for high-quality graphite increases between 10 and 12% per year [15,16], and the price of battery-grade graphite was $500–20,000 per ton, in 2016 [16]. The preparation process of artificial graphite usually requires high temperatures in the (2000–3000) °C range to achieve a well-ordered structure, which accounts for about 50% of the total cost [17]. However, graphite in the residue of recycled batteries already has an ordered structure and does not need to be processed again, thus reducing the manufacturing cost. Hence, recycled material from spent batteries could be an important source of low-cost graphite in the near future. Several investigations have previously reported the obtaining of high-quality graphite from spent lithium-ion batteries. However, usually black mass from smartphones was used, and anode material was previously separated from cathode material to achieve a higher purity of the final graphite.

In this work, we report on the obtaining of high-added value material from black masses of different batteries. Black masses from both Zn/C alkaline and lithium-ion batteries were used as starting materials. Recycled graphites were obtained by a leaching process with an acidic solution. Different conditions which affect the quality of the final graphite were also assessed. A deep characterization of all obtained recycled graphites was also carried out.

## 2. Materials and Methods

### 2.1. Obtaining of Recycled Graphites

Different black masses from spent batteries have been investigated for the recovery of graphite. Two different sieved powders of black mass, including cathode materials, anode materials, and other metal impurities from spent alkaline batteries and lithium-ion batteries were used in the present work as starting materials.

According to previous investigations [18], it is possible to recover metals through acidic leaching of a black mass from spent alkaline and Zn-C batteries, leading to insoluble materials mainly composed of carbon. Thus, 300 g of washed black mass from spent alkaline batteries (BBM) were put in contact with an acid mixture formed by 250 mL of Milli-Q water, 500 mL of 6 M hydrochloric acid (HCl), and 250 mL of H_2_O_2_, and dispersed using mechanical stirring at room temperature for 1 h. Then, the mixture was filtered and the final obtained solid (called C-BBM) was dried at 80 ºC for 24 h. In the case of the black mass from LIBs, spent batteries from both smartphones and vehicles were used. In the latter case and for comparison purposes, two different conditions to obtain high-crystallinity material were assessed. Previous studies have reported that it is possible to leach valuable metals from lithium-ion batteries using a wide range of reagents, such as mineral acids [19,20,21] and organic acids [22,23,24]. Among them, sulfuric acid [25] has successfully been tested for mobilizing the metals from the solid residue to solution, while citric acid has been considered for several years as a potential alternative to inorganic acids for leaching valuable metals from lithium-ion batteries [22]. The spent smartphone batteries were manually dismantled by the separation of the cathode, anode, and aluminum and plastic cases. The final powder, which corresponds to the anodic fraction, was investigated and labelled C-SBM. Finally, 100 g black mass from vehicles were subjected to acidic leaching using (i) 950 mL of 2 M sulfuric acid (H_2_SO_4_) and 50 mL of H_2_O_2_, and (ii) 990 mL of 1.25 M citric acid (C_6_H_8_O_7_) and 10 mL of H_2_O_2_ at 70 °C for 2 h. As previously described, the mixtures were filtered and the final solids (called C-L_1_BM and C-L_2_BM, respectively) were dried at 80 °C for 24 h. In this sense, previous studies have reported that higher metal contents can be recovered using acidic conditions, which would lead to a purer graphite [22,25,26]. A commercial graphite sample (Alfa Aesar, 99.9%) was also investigated for comparison purposes.

The sample notation, starting black mass, and obtaining conditions used in each case are summarized in Table 1.

### 2.2. Characterization

The chemical composition of the starting black masses was determined by X-ray fluorescence (XRF) analysis using a PANalytical Axios wavelength dispersive spectrometer (4 kW). In addition, carbon content was determined by combustion in an induction furnace combustion and infrared detection system.

The structural characterization of the obtained samples was carried out by X-ray diffraction (XRD) using a Bruker D8 Advance diffractometer with Cu K_α_ radiation (λ = 1.5406 Å).

The mean interlayer spacing, d_002_, was determined using the Bragg′s equation (Equation (1)). The graphitization degree (g) was determined using the equation proposed by Maire and Mering [27] (Equation (2)), 0.3440 is the interlayer spacing of the fully nongraphitized carbon (nm), 0.3354 is the interlayer spacing of the ideal graphite crystallite, and d(002) is the interlayer spacing derived from XRD (nm). 0.3440 nm represents a certain specific structure according to Franklin [28,29] who considered it as an interlayer spacing of the non-graphitic carbons. The stacking height (L_c_) was evaluated using the Scherrer equation (Equation (3)). These parameters are commonly used to evaluate the degree of structural order of the carbonaceous materials.
(1)$$d_{002}\left(\mathrm{nm}\right)=\frac{\lambda }{2\sin\theta }$$
(2)$$g\left(\%\right)=\frac{0.3440-d_{002}}{0.3440-0.3354}\cdot 100$$
(3)$$L_{c}\left(\mathrm{nm}\right)=\frac{k\cdot \lambda }{\beta_{002}\cdot \cos\theta_{002}}$$

In these equations, λ is the X-ray wavelength, θ is the Bragg angle corresponding to the diffraction maximum, k is the Scherrer constant (k = 0.94), and β is the full width at half maximum (FWHM) of the diffraction peak.

The morphology of the samples was investigated with a FEI Inspect S scanning electron microscope (SEM), while its chemical composition and elemental spatial distribution were assessed by energy dispersive X-ray microanalysis (EDX, Bruker Quantax) in a Leica 440 Steroscan SEM. An accelerating voltage of 20 kV and a beam current of 1.5 nA were used for EDX measurements.

Micro-Raman measurements were carried out at room temperature in a Horiba Jovin-Ybon LabRAM HR800 system. The samples were excited by a 633 nm He-Ne laser on an Olympus BX41 confocal microscope with a 100× objective. The spectral resolution of the system used was ~1 cm^−1^. Laser power density was carefully adjusted in order to avoid heating or irradiation effects. Spectra shown in this work were taken under the same experimental conditions for comparison purposes.

The electrochemical tests were performed in Swagelok-type cells assembled in an Ar-filled dry box, using Li metal as the counter electrode. A Whatman GF/D borosilicate glass fiber sheet was saturated in the electrolyte. The electrolyte was purchased from Sigma-Aldrich and consists of a lithium hexafluorophosphate solution in ethylene carbonate and dimethyl carbonate, 1.0 M LiPF_6_ in EC/DMC = 50/50 (*v*/*v*). The recycled graphite (90 wt%) was mixed with 5 wt% of carbon SP (black carbon, MMM, Belgium) and 5 wt% of sodium alginate in deionized water. Na alginate is a green binder, non-toxic, inexpensive, easy to process, and with a high drying rate. Moreover, it can be dissolved in water and easily disposed at the end of the battery life. The sodium alginate, carbon SP, and graphite were dispersed in water and cast on a 25 μm copper foil using a doctor-blade technique with a wet gap of 100 μm. The electrode was then heated at 100 °C using a compact programmable film coater with vacuum chuck, film applicator, and drying cover (LITH-TMJ200). The resulting electrodes had an active material loading of around 3–10 mg/cm^2^. Electrochemical testing was carried out by using a Biologic 815 potentiostat. Galvanostatic charge and discharge tests were conducted in a potential window between 2 and 0.01 V at different current densities. The applied current densities were normalized to the mass of graphite on the electrode.

## 3. Results and Discussion

### 3.1. X-ray Fluorescence (XRF)

The starting black masses were dried at 80 °C and sieved to obtain representative fractions with particle size below 80 μm. Its chemical composition was determined by X-ray fluorescence (XRF). The obtained results are given in Table 2. The group “Others” corresponds to elements found in concentrations below 0.01 wt.% as well as to light elements such as Li, O, N, and F, including C.

In the case of the black mass from spent LIBs, some chemical elements can be associated with a particular component of the battery such as aluminum to the cathode current collector; copper to the anode current collector, and iron to the casing particles. Other metals such as cobalt, nickel, and manganese were also detected and can be associated to the lithium metal oxides.

In addition, carbon content was determined by combustion in an induction furnace and infrared detection, as specified in the last row of Table 2.

### 3.2. X-ray Diffraction (XRD)

Structural characterization of the final obtained solids and the commercial sample was initially carried out by XRD. XRD patterns of the investigated samples are shown in Figure 1. Data have been plotted in log scale in order to better visualize low intensity diffraction maxima.

In all cases, the more intense diffraction maxima were indexed to the carbon-graphite phase with hexagonal structure and space group P6_3_/mmc (JCPDS 00-056-0159). The narrower and higher peak centered at around 26.5° can be attributed to the reflection in the (002) plane of aromatic layers. Other peaks centered at 42°, 44°, and 54°, which correspond to the (100), (101), and (004) planes of graphite carbon, can also be observed. The (002) peak is attributed to the orientation of the aromatic ring carbon reticulated layers in three-dimensional arrangement. The (100) peak is related to the degree of condensation of the aromatic ring (i.e., the size of the carbon slice of the aromatic layer [19]). The sharper and the higher the (002) and (100) peaks are, the better the orientation and the larger the size of the aromatic layer slice, respectively.

A comparative analysis of the structural data of the final obtained materials is summarized in Table 3. In all cases, the calculated d_002_ of the recycled graphites were very similar to that of ideal graphite (0.3354 nm) with d_002_ values < 0.3440 nm. Therefore, they can be classified within graphitic materials [28]. In addition, graphitization degree was higher than 87% for all obtained samples. These results suggest that the final obtained samples are high-quality recycled graphites. Finally, the calculated L_c_ values increase for decreasing d_002_ interlayer spacing.

Besides graphite-related diffraction maxima, XRD patterns of the C-L_2_BM material also show much weaker peaks that can be attributed to secondary phases, in particular to Al_2_O_3_ (JCPDS 00-46-1212) and Cu (JCPDS 00-04-0836). As previously mentioned, these impurities can be attributed to cathode and anode current collector, respectively. No more impurities were detected within the sensibility of these measurements.

### 3.3. Scanning Electron Microscopy (SEM)

Microstructural characterization of the obtained samples was carried out by scanning electron microscopy (SEM). Figure 2 shows representative SEM images of the investigated materials. In all cases, agglomerates of densely packed C layers of different sizes were observed. Interspersed among such agglomerates, smaller particles can also be appreciated. However, small differences can be appreciated. C-BBM sample exhibits less packed layers than the other analyzed samples, in which the size varies between 10 and 200 μm approximately with a flake-like morphology. A similar morphology can be appreciated in the samples obtained from black masses of spent lithium-ion batteries. In these cases, more rounded particle shape was found, with agglomerates sizes in the range of (2–40) μm, (5–20) μm, and (10–30) μm for the C-SBM, C-L_1_BM, and C-L_2_BM samples, respectively.

### 3.4. EDX Microanalyses

The chemical composition of the samples, as well as their corresponding elemental spatial distribution, were investigated by energy dispersive X-ray microanalysis. In all cases, EDX microanalyses (Table 4, and Figure 3) reveal that C is the main element in the samples, although the presence of some impurities in low concentrations is clearly detected. Spectra are represented in log scale in order to visualize elements present in lower concentrations more clearly.

In the case of the final materials obtained from black masses of spent lithium-ion batteries, the concentration of fluorine is high. This is due to the presence of the electrolyte in the starting black masses, since electrolyte solutions based on fluorinated solvents are commonly used in LIBs because of their extraordinary electrochemical stability [30,31]. Moreover, copper and silicon were also detected, in good agreement with the obtained results from XRD measurements. Both elements are related to the electrical components and the case of the batteries.

The spatial distribution of the elements detected in the different samples are shown in Figure 4, Figure 5, Figure 6 and Figure 7.

Carbon distribution is homogeneous throughout the whole material in all the investigated samples. No clear correlation seems to exist between the detected elements in the case of the C-BBM sample (Figure 4).

The elements detected in the C-SBM sample (Figure 5) appear to be quite homogeneously distributed, with the exception of P and Cu. The corresponding mappings reveal that the distributions of these two elements are spatially correlated.

In the case of the C-L_1_BM material (Figure 6), a clear correlation between the distribution of Si and O, as well as between the distribution of Al and O can be appreciated, which strongly suggest the existence of alumina and silica particles in the sample. The absence of diffraction maxima corresponding to silica and alumina can be attributed to the low concentration of these phases and/or their low crystallinity. Although Cu is detected in the EDX spectra, the corresponding mappings revealed no particular feature irrespective of the area sampled or the SEM magnification used, which strongly suggests that the mentioned signal comes from the SEM sample holder.

Finally, the spatial distribution of Al is strongly correlated with that of O in the case of the C-L_2_BM sample (Figure 7), evidencing the existence of alumina particles. Diffraction maxima related to alumina were in fact detected in the XRD pattern of this sample. In addition, the spatial distribution of Cu is not correlated with that of O. Metallic Cu particles with sizes ranging approximately between 1 and 10 µm can be observed in the corresponding mapping, although their surface is probably slightly oxidized due to contact with air. In this sample, Cu signal clearly does not stem from the SEM holder. This result is also in good agreement with our XRD measurements, where Cu diffraction maxima were found.

### 3.5. Micro-Raman Spectroscopy

All the obtained samples were also characterized by micro-Raman spectroscopy. This non-destructive spectroscopic technique is particularly suitable for the characterization of C-based compounds, providing information regarding its morphology, defect structure, and graphitization degree, and is an important tool for the characterization of the different carbon allotropes due to its sensitivity to structural changes [32,33]. While XRD provides long-range structural information about graphite [34] and is not a spatially resolved technique—meaning that somehow averages the structural characteristics of graphite particles with different size, morphology, and defect structure—micro-Raman spectroscopy is a spatially resolved technique that provides short-range structural information of individual particles, being both techniques complementary. Hence, the combined use of both non-destructive methods constitutes a unique tool for a consistent structural assessment of our materials. Representative spectra of each sample are shown in Figure 8. Several bands related to different vibrations of C atoms in graphite can be observed. The relative intensities of the mentioned bands were found to depend, to a certain extent, on the size of the particle or flake probed by the laser beam. This is specially the case for the commercial sample. Their peak positions and corresponding assignments are shown in Table 5.

Some of these bands are not evident but found by deconvolution of the experimental data to a sum of Lorentzian profiles. The D peak is a defect-activated band associated to the breathing modes of six-atom rings K-point phonons (A_1g_ symmetry). The D´ band is also related to defects, since it originates from intravalley one-phonon double resonance Raman processes involving one longitudinal optical phonon near the Γ point of the Brillouin zone (BZ) and one defect. These modes are not Raman active in first order Raman scattering of perfect crystals, since they are not zone-center modes, but become Raman active in defective graphitic materials owing to defect-induced double resonance Raman scattering processes involving the electronic π–π* transitions [35]. The G band is a doubly degenerate phonon mode (E_2g_ symmetry) at the BZ center that is due to the bond stretching vibrations of all pairs of sp^2^ atoms in both rings and chains of carbon networks [36,37]. The 2D band corresponds to the harmonic (second order Raman scattering) of an in-plane transverse optical (TO) mode close to the zone boundary K point [35]. This band appears as a doublet due to the splitting′s of the π and π* electronic states, owing to the interactions between the successive layer planes.

Some bands related to the second order Raman spectrum were also detected. In the case of the C-BBM sample, three weak bands at 2470, 2922, and 3241 cm^−1^, assigned to the combinations D + D″, D + G, and to the harmonic 2D′, respectively, were found. In the other cases only two weak bands at 2456, 2458, and 2467 cm^−1^, assigned to the combination D + D″, and 3242, 3240, and 3241 cm^−1^ assigned to the harmonic 2D′, were detected for the C-SBM, C-L_1_BM, and C-L_2_BM, respectively. D″ corresponds to a phonon belonging to the in-plane longitudinal acoustic (LA) branch close to the K point [38] and D′ corresponds to a phonon of the in-plane longitudinal optical (LO) branch close to the zone center (Г point) [39]. Moreover, in the case of the C-BBM sample, a low-intensity band can be appreciated at about 1155 cm^−1^ (D* band). The origin of this band is still controversial, since it has been observed in graphene oxide samples with different chemical compositions [40,41], nanocrystalline diamond [42] and few-layer wrinkled graphene [22], amongst other carbon materials. Some authors attributed the D* peak to sp^3^ carbons in amorphous or disordered graphitic lattices [20,22], while others assign this peak to C=C stretching and CH wagging modes of trans-polyacetylene and not to sp^3^ carbons [21], which is unlikely in the present case.

The intensity ratio of the (I_D_/I_G_) bands reflects the order degree of the graphite [19], the intensity of the D′ peak is proportional to the amount of defects [43], while the full width at half maximum (FWHM) of the G band is known to reflect the surface crystallinity of the carbon material and decreases for increasing crystallite size [44]. The ranges of variation of the intensity ratios I_D_/I_G_ and I_D´_/I_G_ as well as the FWHM of the G band measured for each sample are summarized in Table 6. According to the obtained results, the relative intensities of the C-BBM sample can be considered representative of a crystalline graphite with a rather high density of defects, as evidenced by the higher (I_D_/I_G_) and (I_D´_/I_G_) ratios, as well as the higher FWMH of the G band, as compared with that measured in the other samples. Furthermore, the relative intensities of the mentioned bands for the C-SBM, C-L_1_BM and C-L_2_BM samples can be considered representative of good quality crystalline graphites, with a rather low density of defects. These results are in good agreement with that obtained from XRD measurements, where the C-BBM sample exhibit a lower graphitization degree (g, %) than the other analyzed samples.

On the other hand, micro-Raman measurements were also carried out in small particles, (1–4) µm in size, of the C-L_2_BM sample, showing a different appearance in SEM micrographs.

Some examples are shown in (Figure 9). Sometimes, the laser spot is wide enough to excite the Raman response of both the particle and the surrounding graphite, as revealed by the appearance of the D (1333 cm^−1^), G (1579 cm^−1^), and 2D (2682 cm^−1^) bands. Raman peaks corresponding to these particles appear centered at about 88, 144, 213, 422, 523, and 621 cm^−1^ and can be all attributed to Cu_2_O [45,46,47], evidencing the existence of particles of this oxide or Cu particles whose surface has been oxidized as a result of prolonged exposure to ambient air. It should be mentioned that, due to symmetry reasons, Cu is Raman inactive. Hence, detection of metallic Cu particles is not possible by using this spectroscopic technique. According to group theory, Cu_2_O has six zone-center optical phonon modes which are classified as: Γ = F_2g_ + 2F_1u_ + F_2u_ +E_u_ + A_2u_. The F_2g_ mode (~515 cm^−1^) is Raman-active and the two F_1u_ modes (~144 and ~630 cm^−1^) are IR active for a perfect Cu_2_O lattice [45,46,47]. However, Cu_2_O is frequently nonstoichiometric and defects (including O and Cu vacancies) may induce a breakdown of the selection rules and activate Raman inactive modes [46,47]. In addition, multiphonon processes can also be observed in our Raman spectra. In fact, the 213 cm^−1^ peak is attributed to a 2E_u_ overtone while the mode at ~422 cm^−1^ is due to a multiphonon process [46,47].

The obtained results indicate that the recycled graphite showing the best crystallinity and lowest defect concentration is that of sample C-L_1_BM. The suitability of that sample for Li ion batteries anodes was then assessed. It should be mentioned that contrary to previous studies where toxic and harmful polyvinylidene fluoride (PVDF) was used, a green, non-toxic, and inexpensive binder, namely sodium alginate, was employed in the present case. The battery was first tested at a current density of 74 mA·g^−1^ (0.2 C) for 20 cycles in order to check the initial capacity and its stability. Figure 10a shows charge-discharge profiles at 74 mA g^−1^ (0.2 C) between 2 and 0.001 V. From the first charge/discharge curve, C-L_1_BM delivers a discharge capacity of 433 mA·h·g^−1^ and corresponding charge capacity was of 182 mA h g^−1^, which shows an irreversible capacity of nearly 251 mA h g^−1^. This irreversible capacity is due to the formation of the solid electrolyte interface (SEI) in the first lithium intercalation process [48]. In the second cycle, the discharge capacity decreases and the charge increases, while in the fifth cycle the capacity achieved was 325 mA h g^−1^ and the coulomb efficiency reached 100%. Figure 10b shows the first charge-discharge curves at three different current densities, which are similar to that reported previously in the literature for lithium-graphite batteries [49].

Cycling performance at 0.2 C is shown in Figure 11a. After 20 cycles, the specific capacity stabilizes at 366 mA g^−1^, which is very close to the theoretical specific capacity of graphite (372 mA g^−1^), and the coulombic efficiency remains above 98%. The rate performance of the electrode can be observed in Figure 11b. Specific capacities of 370, 186, and 62 mA h g^−1^ are respectively retained after 20 cycles at 500, 1000, and 2000 mA·g^−1^ (0.5 C, 1 C, and 2 C). When current density is switched back at 500 mA·g^−1^, a specific capacity of 280 mA g^−1^ is measured after another 20 cycles. Irrespective of the current density, the coulombic efficiency remains above 93%. These results can be considered very promising taking into account the theoretical specific capacity of graphite, the binder employed, and the recycled nature of the material used.

## 4. Conclusions

High-quality recycled graphites have been obtained from black masses, a common waste product generated in the recycling of spent batteries. Black masses from spent Zn/C alkaline and from lithium-ion batteries were assessed. Despite their different starting compositions, acid leaching was successfully used to separate the metal content present in the mentioned black masses. XRD patterns of the investigated samples reveal carbon-graphite as the principal crystalline phase. Only minor amounts of second phases were detected in some samples. Materials obtained from different types of black masses show uneven morphologies. SEM micrographs show densely packed agglomerates in the case of the graphite obtained from alkaline batteries, while rounded particles were observed for the materials obtained from LIBs black masses. EDX spectra and elemental mappings reveal that the obtained materials are mainly composed by carbon, with the exception of minor impurities, which indicate that acidic leaching was successful. Raman spectra of all the investigated samples mainly exhibit bands than can be attributed to crystalline graphite. In the case of the graphite obtained from the black mass of spent Zn/C alkaline batteries, the measured relative intensities (I_D_/I_G_) and (I_D_′/I_G_) bands, as well as the higher FWMH of the G band can be considered representative of crystalline graphite with a rather high density of defects. However, spectra from graphites obtained from black masses of spent LIBs can be considered representative of a good quality crystalline material with a rather low density of defects, as evidenced by the lower (I_D_/I_G_) and (I_D_′/I_G_) ratios and the lower FWMH of the G band. These results evidence the obtaining of a high-added value material from different kinds of spent batteries through acid leaching. Moreover, the suitability of the sample showing the best crystallinity for Li ion batteries anodes was tested. After 20 cycles at 0.2 C, the specific capacity stabilizes at 366 mA h g^−1^, which is very close to the theoretical specific capacity of graphite (372 mA h g^−1^). Specific capacities of 370, 186, and 62 mA h g^−1^ are respectively retained after 20 cycles at 0.5 C, 1 C, and 2 C. These results can be considered very promising taking into account the recycled nature of the material used.

## Figures and Tables

**Figure 1 materials-15-03246-f001:**
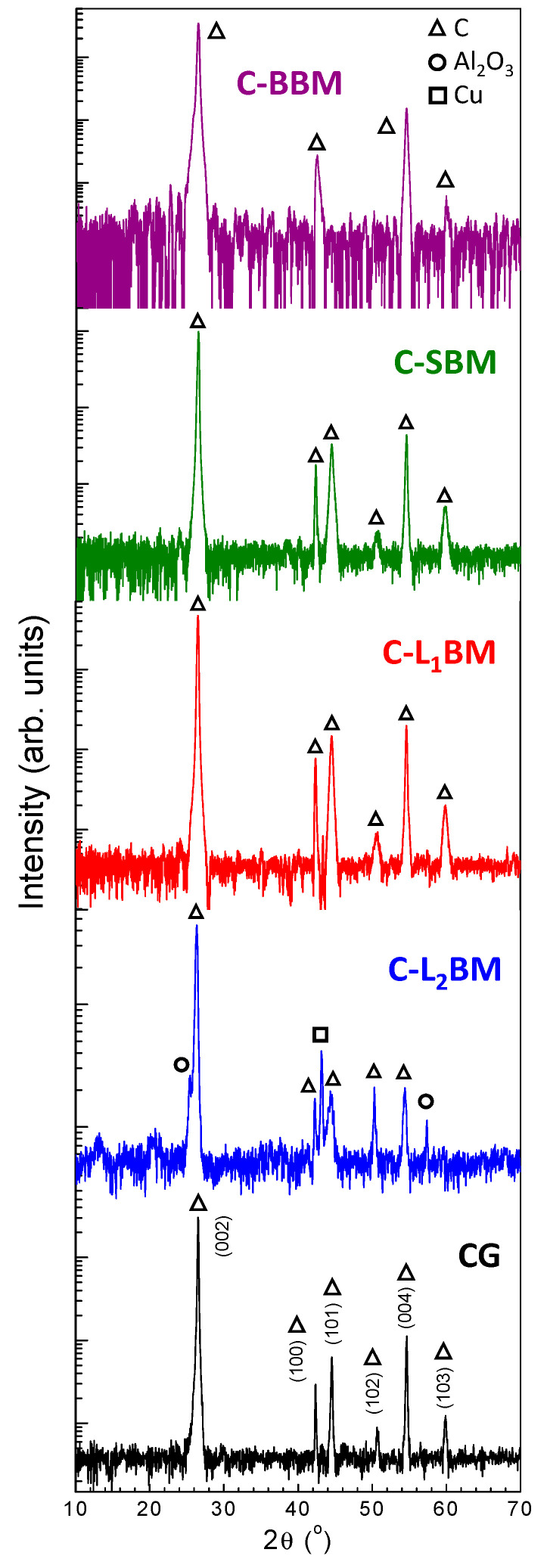
XRD patterns of the obtained final solids.

**Figure 2 materials-15-03246-f002:**
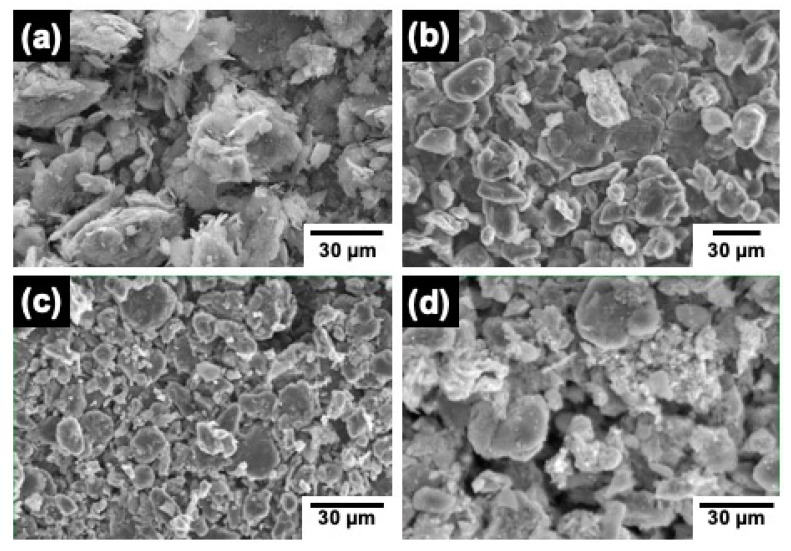
SEM micrographs representative of the investigated samples: (**a**) C-BBM, (**b**) C-SBM, (**c**) C-L_1_BM, and (**d**) C-L_2_BM.

**Figure 3 materials-15-03246-f003:**
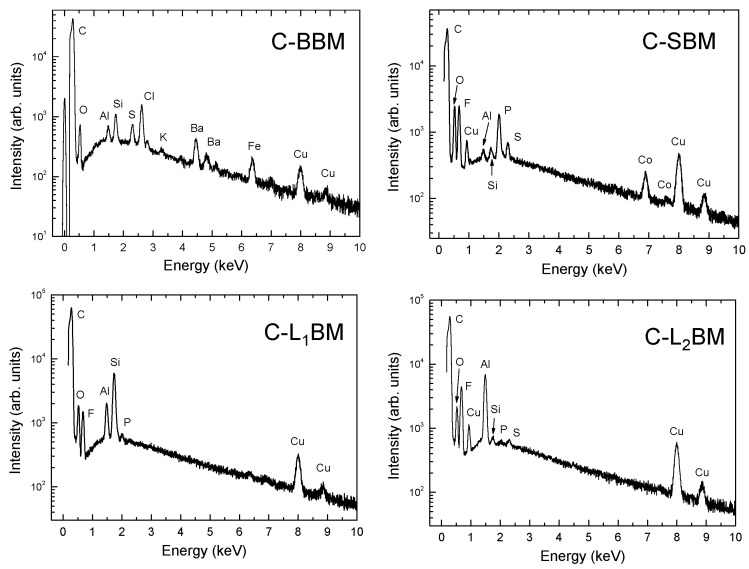
Representative EDX spectra of the investigated graphite samples.

**Figure 4 materials-15-03246-f004:**
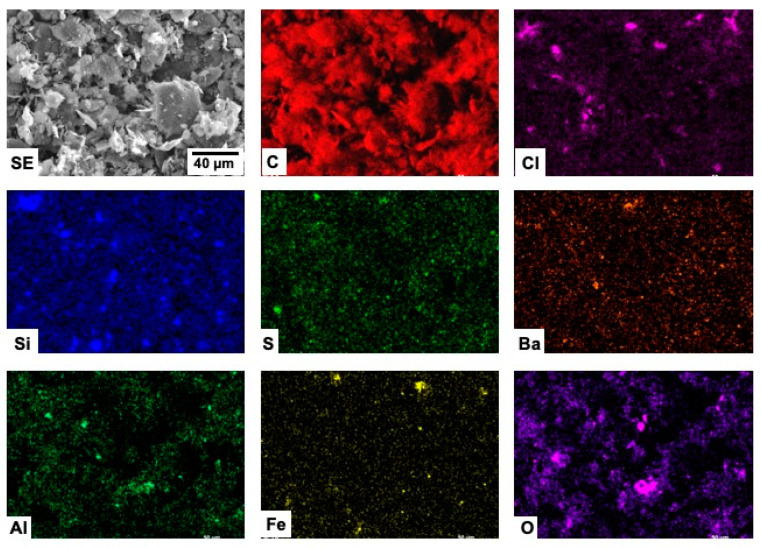
SEM-EDX mappings from sample C-BBM.

**Figure 5 materials-15-03246-f005:**
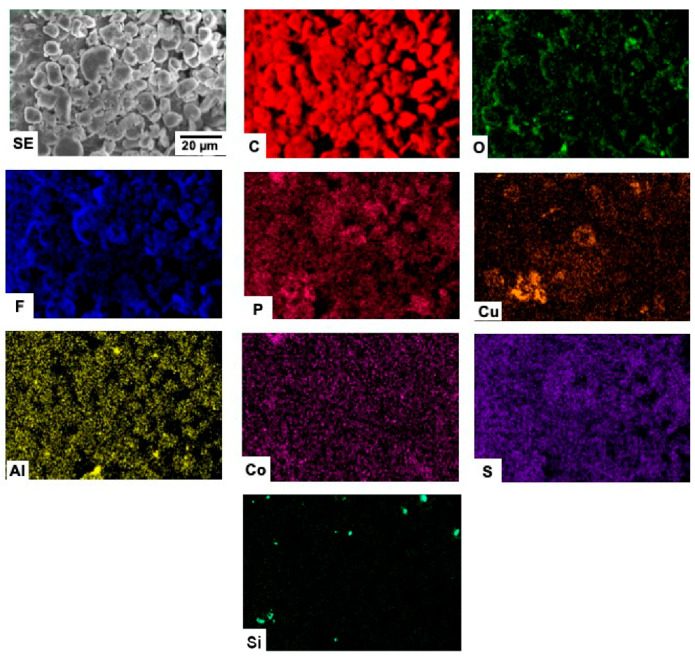
SEM-EDX mappings from sample C-SBM.

**Figure 6 materials-15-03246-f006:**
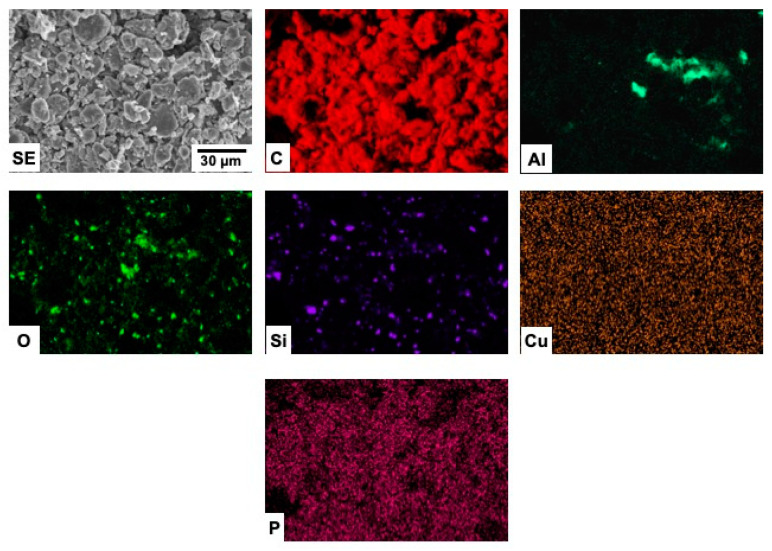
SEM-EDX mappings from sample C-L_1_BM.

**Figure 7 materials-15-03246-f007:**
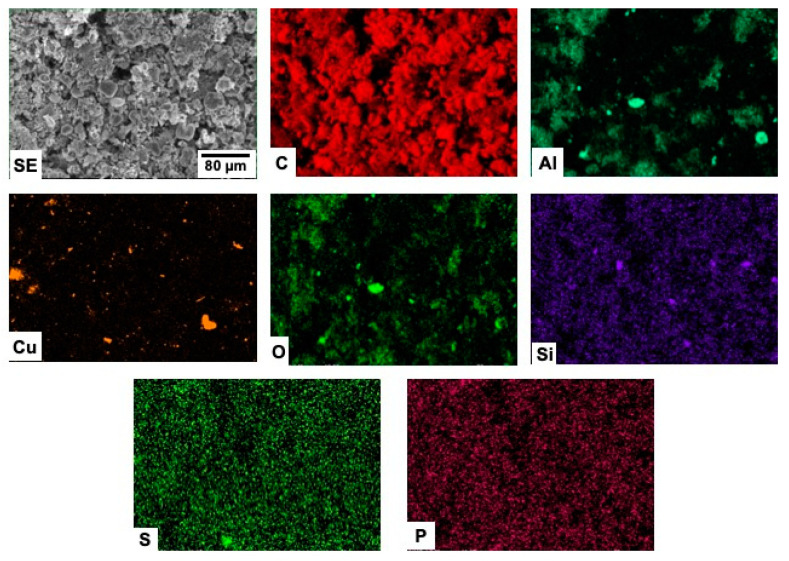
SEM-EDX mappings from sample C-L_2_BM.

**Figure 8 materials-15-03246-f008:**
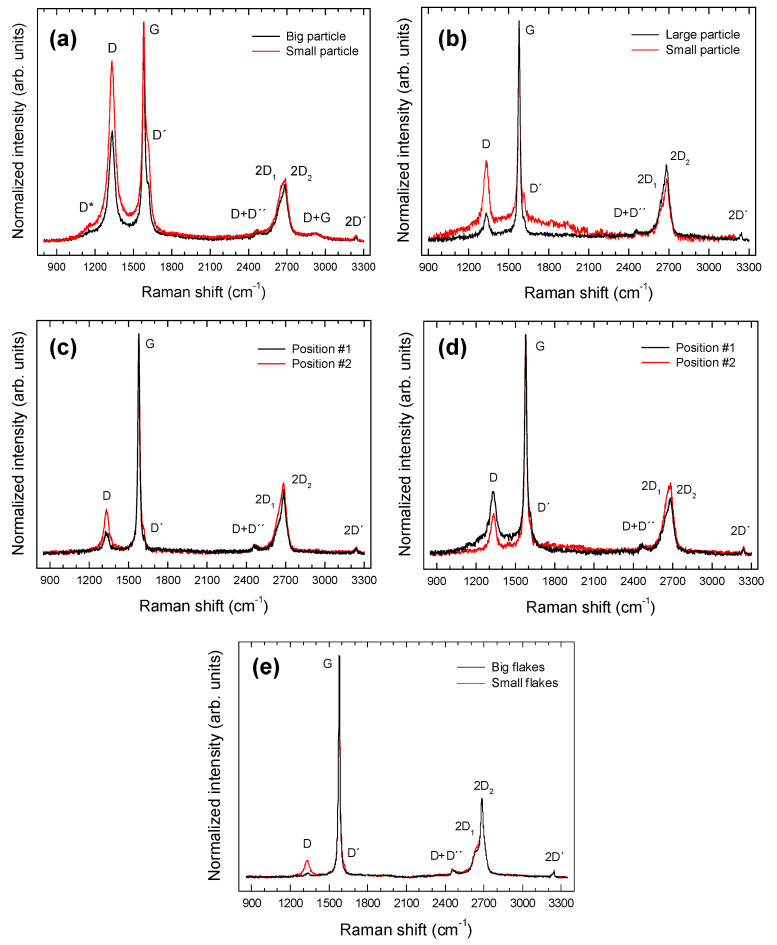
Representative Raman spectra from the (**a**) C-BBM, (**b**) C-SBM, (**c**) C-L_1_BM, (**d**) C-L_2_BM, and (**e**) commercial graphite samples.

**Figure 9 materials-15-03246-f009:**
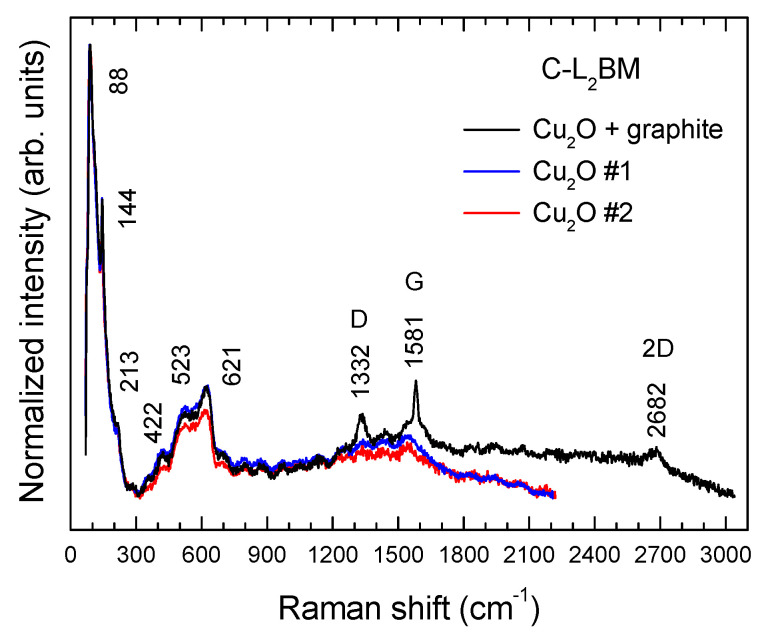
Raman spectra of small Cu_2_O particles from the C-L_2_BM sample.

**Figure 10 materials-15-03246-f010:**
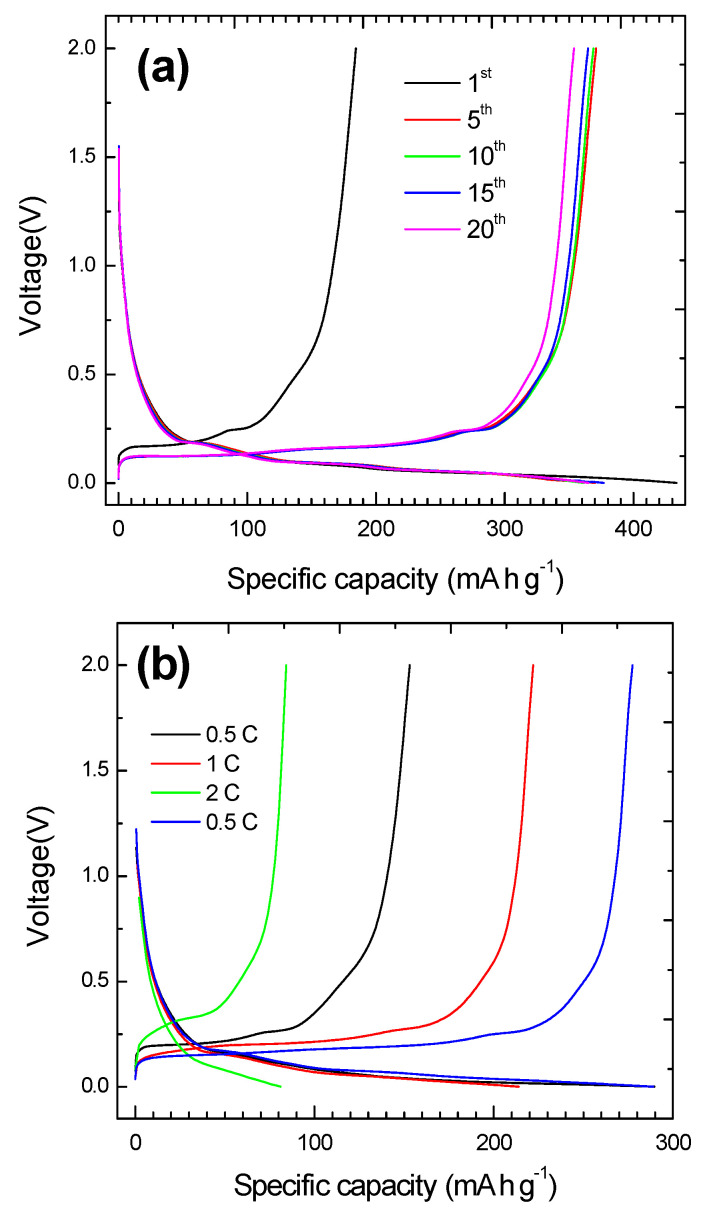
(**a**) Charge/discharge curves at 0.2 C of a Li-ion battery using recycled graphite CL_1_BM as anode material. (**b**) First charge/discharge curves at different current densities.

**Figure 11 materials-15-03246-f011:**
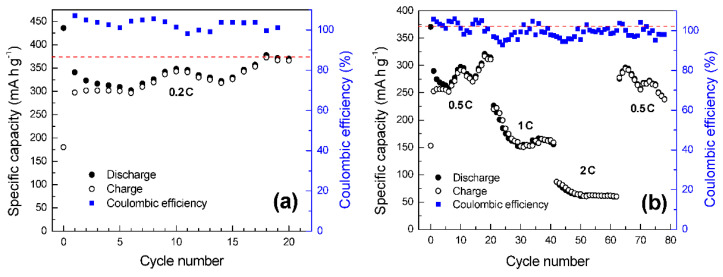
(**a**) Cycling performance at a current density of 74 mA g^−1^ (0.2C) of a Li-ion battery using recycled graphite CL_1_BM as anode material. The red dashed line indicates the theoretical capacity of graphite (372 mA h g^−1^). (**b**) Rate performance.

**Table 1 materials-15-03246-t001:** Samples notation and obtaining conditions for the investigated samples.

Sample Notation	Black Mass	Leaching Conditions
C-BBM	Black mass from spent Zn/C alkaline batteries	6 M HCl/H_2_O_2_ (25% *v*/*v*)
C-SBM	Black mass from spent smartphone lithium-ion batteries	Untreated
C-L_1_BM	Black mass from spent vehicles lithium-ion batteries	2 M H_2_SO_4_/H_2_O_2_ (5 % *v*/*v*)
C-L_2_BM	Black mass from spent vehicles lithium-ion batteries	1.25 M C_6_H_8_O_7_/H_2_O_2_ (1 % *v*/*v*)

**Table 2 materials-15-03246-t002:** Chemical composition (%wt) of the starting black masses. The carbon content appearing in the last row is that determined by combustion in an induction furnace and infrared detection.

Compound	C-BBM	C-SBM	C-LBM
Na_2_O	7.75	0.06	-
MgO	0.13	-	-
Al_2_O_3_	0.46	0.04	2.25
SiO_2_	0.19	0.08	0.06
P_2_O_5_	0.99	0.75	1.08
SO_3_	0.64	0.15	0.21
Cl	1.76	-	0.04
K_2_O	6.70	-	-
CaO	0.36	-	0.07
TiO_2_	0.16	-	0.01
MnO	41.40	0.04	6.45
Fe_2_O_3_	1.42	0.02	0.03
Co_3_O_4_	0.03	0.20	4.43
NiO	0.18	-	6.84
CuO	0.08	0.94	1.17
ZnO	26.88	-	0.95
SrO	0.04	-	-
ZrO_2_	0.01	-	0.14
CdO	0.01	-	-
SnO_2_	0.03	-	0.01
PbO	0.04	-	-
Others	10.74	97.92	76.26
C	5.42	85.59	33.87

**Table 3 materials-15-03246-t003:** Structural parameters calculated for the graphite samples. CG stands for commercial graphite.

Sample	d_002_ (nm)	g (%)	L_c_ (nm)
C-BBM	0.3363	87	30.47
C-SBM	0.3358	92	40.62
C-L_1_BM	0.3356	96	44.90
C-L_2_BM	0.3360	91	32.81
CG	0.3355	97	45.14

**Table 4 materials-15-03246-t004:** Quantification of EDX analyses (normalized %wt) carried out in the final graphite samples.

Element	C-BBM	C-SBM	C-L_1_BM	C-L_2_BM
Carbon	80.97	56.4	73.65	55.4
Oxygen	14.77	15.1	14.86	9.523
Barium	1.29	-	-	-
Chlorine	1.20	-	-	-
Iron	0.49	-	-	-
Silicon	0.427	0.1	1.88	0.054
Sulfur	0.279	0.2	-	0.079
Aluminum	0.191	0.1	0.81	3.02
Potassium	0.082	-	-	-
Copper	0.3	2.4	0.8	3.81
Fluorine	-	54	7.85	28
Cobalt	-	0.59	-	-
Phosphorus	-	1.07	0.045	0.066

**Table 5 materials-15-03246-t005:** Raman bands positions (cm^−1^) observed in the investigated samples and their assignments.

Band	C-BBM	C-SBM	C-L_1_BM	C-L_2_BM	CG
D*	~1155	n.o.	n.o.	n.o.	n.o.
D	1332	1332	1332	1333	1330
G	1581	1579	1579	1578	1580
D′	1615	1617	1619	1616	1616
D + D″	~2470	~2456	~2458	~2467	2467
2D_1_	2647	2646	2646	2653	2654
2D_2_	2686	2686	2686	2685	2687
D + G	2922	n.o.	n.o.	n.o.	n.o.
2D′	3241	3242	3240	3241	3241

(n.o. ≡ not observed).

**Table 6 materials-15-03246-t006:** Variation ranges of the intensity ratios I_D_/I_G_ and I_D´_/I_G_ and FWHM of the G band for all the samples investigated.

Intensity Ratio/Sample	C-BBM	C-SBM	C-L_1_BM	C-L_2_BM	CG
(I_D_/I_G_)	0.50–0.82	0.1–0.3	0.11–0.21	0.19–0.29	0.02–0.27
(I_D´_/I_G_)	0.27–0.48	0.03–0.09	0.03–0.05	0.04–0.08	0.01–0.08
FWMH (cm^−1^)	19.6–24.9	16.7–17.4	16.5–17.7	19–20	13.6–18.4

## Data Availability

Not applicable.

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
