# Peer review of "Obtaining and Characterization of Highly Crystalline Recycled Graphites from Different Types of Spent Batteries"

_materials, 2022, doi:10.3390/ma15093246_

Round 1
Reviewer 1 Report
- Line 74: final obtained solid (called C-BBM) was dried at 80 ºC for 24 h. In the case of the black. A vacuum drying oven is preferred.
- Line 131: corresponds to elements found in concentrations below 0.01 wt.% as well as to light elements such as Li, O, N and F, including C.
- Line 324: Figure 9. Raman spectra of small Cu2O particles from the C-L2BM sample. It would be appreciated if you would show XPS spectra of small Cu2O particles from the C-L2BM sample.
Reviewer 2 Report
Manuscript materials-1669878 “Obtaining and characterization of high-quality recycled graphites from different types of spent batteries” of L. Alcaraz, C. Diaz-Guerra, and F. A. Lopez is devoted to one of the important problems of modern ecology, namely the reuse of materials from used devices containing lithium-ion batteries. They involve the reuse of graphite-containing materials after leaching with acidic reagents.
I would like to express my observations and comments that arose after reading this manuscript.
1 The manuscript does not clearly state the criteria that graphite recycled from used devices must meet. Their absence does not allow assessing the effectiveness of material treatment. Not all high-carbon materials can be reused in accumulators.
- The manuscript does not clearly indicate what criteria the authors used to select reagents for acidic purification?
- The C-SBM sample without acid leaching has structural parameters that do not differ at all from those corresponding to the treated materials (Table 3). I have an opinion that it is possible not to process at all. However, it is possible that some details of preparation of this sample are not specified in the manuscript.
- I would not agree with the authors that equation 2 is due to Dr. R. Franklin (she is an outstanding scientist!), I think that the equation was first proposed by Maire and Mering. Equation 2 is empirical and cannot be used as one of the main structural parameters of the studied graphite samples. For a correct determination of the degree of graphitization, it is necessary to use peaks 110 and 112. These peaks in Figure 1 are not shown. More fundamental references of literature can be used to explain the diffraction results.
- There is a typo in Equation 2.
- I did not find Equation 4.
- There is a discrepancy between Table 4 and Figure 3 in the content of the Copper element, which corresponds to sample C-BBM.
- As a wish, it would be appropriate to present some results of electrochemical tests of cells prepared using purified materials.
It seems to me that this manuscript can be published after revision, adding the missing criteria and correcting the typographical errors.

Reviewer 3 Report
See attached file.

Reviewer 4 Report
With the rapidly commercial application of lithium-ion batteries in electric vehicles, a huge number of spent batteries is decommissioning soon. More efficient recycling approaches will bring more environmental and economic benefits. Graphite was discarded in the recycling process, but now, it attracts more and more attention. In this work, authors recycled graphite from three different spent battery streams and characterized their physical properties. I would like to recommend the paper to be published in Materials after authors address the following questions:
- The title should be changed. The recycled graphite is not high-quality. The obtained graphite still has impurities. The high-quality should be deleted from the title.
- In the introduction section, authors said recycled graphite is a high added-value material and it can reduce the cost of graphite. I would like to know if authors have any information about the cost of recycled graphite. Is it cheaper than commercial graphite?
- Authors only introduced recycling graphite by mechanical methods, which is not accurate and misleading. Ma, Xiaotu, et al. "High-performance graphite recovered from spent lithium-ion batteries." ACS Sustainable Chemistry & Engineering24 (2019): 19732-19738. has introduced recycled graphite from the hydrometallurgical process.
- In Table 2, the C content of C-SBM is extremely high. Compared to 33.87 wt% in C-LBM, it is ~2.5 times in C-SBM, which both are Lithium-ion batteries. Based on my knowledge, I don’t think the carbon-based materials will take over 80wt% in Li-ion batteries.
- In Figure 1, authors marked Cu peaks in C-L2BM materials. C-L2BM is after leaching. Based on the solid/liquid ratio in the experiment section, Cu should be easily dissolved in the acid. The Cu peaks are close to the Al2O3 peaks. In the standard peaks, Al2O3 peaks are at 43.36 and 52.55, which is close to Cu's peaks at 43.30 and 50.48, which are recognized as Al2O3 peaks in Ma et al ‘s paper.
- Where is the Si from? I’m not 100% sure if Zn/C alkaline has Si, but I’m pretty sure that this generation of Li-ion batteries should not have Si.
- In Figure 3, all EDX spectra have a silicon peak. The strange thing is that C-SBM has a lower content of Si than C-L1BM. As we all know, the Si is not dissolved in sulfuric acid and critic acid. Why the content of Si is changed before and after leaching.
- Figure 5 does not have a Si mapping, which has a Si peak in Figure 3.
- It is better to add commercial graphite as a reference in the XRD and Raman test.

Round 2
Reviewer 2 Report
I thank the authors for taking into account the comments and additions made, but I still have some observations.
Electrochemical tests give rise to a number of remarks.
- Carbon materials have porosity or large specific surface, these factors lead to water adsorption by these materials. Therefore, drying the electrode at room temperature and without vacuum is insufficient for electrode preparation.
- Since the manufacturer of the electrolyte is not indicated, I can assume that the electrolyte is homemade. What was the moisture content of the electrolyte?
- The manuscript does not provide charge-discharge profiles. And therefore it is impossible to explain the sharp fluctuations in capacitance during testing.
- How was the zero cycle capacity measured?
- The Coulomb efficiency of the first cycle is always less than 100% (part of the charge is irreversibly spent on the formation of SEI). How can authors explain the Coulomb efficiency for first cycle above 100%?
- Coulomb efficiency does not have a smooth dependence, and side reactions are most likely observed.
- Caption to figure 10. The current density is measured in A per unit area or mass, while the signature indicates the unit of specific gravimetric electrochemical capacity - mA h /g.
0.2 C for graphite (per 1 g) will correspond to 74.4 mA, since 1 C for graphite is equal to 372 mA. The same remark applies to the text of the manuscript on p. 13, lines 389-390.
- Misprinting was made in equation 2, in the numerator instead of "0.334" it is necessary to write "0.344", as indicated in the text of the manuscript.
- The XRD spectra are presented in such a way that their interpretation is very difficult or even impossible. For example, it is possible to use logarithmic scale for peak intensity.
It seems to me that this manuscript can be published after full revision of electrochemical part.
Reviewer 3 Report
I feel satisfied with the correction made by authors.
Author Response
We would like to thank the reviewer his/her comments.
Reviewer 4 Report
Having read through the responses to my previously raised review points, I am grateful for the diligent and thorough way that you have worked through and attended to all of the replies. I thank you for the courtesy of taking on board the feedback and hope that it was considered useful by the writing team. My recommendation now is to publish this article without reservation. I raised my concerns in the previous round of comments and I am very satisfied with the way that these have either been addressed or defended. I have nothing further to add.
Author Response

(The authors gave the same response as above.)
